# Combined Reaction System for NH_3_ Decomposition and CO_2_ Methanation Using Hydrogen Permeable Membrane Reactor in 1D Model Analysis

**DOI:** 10.3390/membranes14120273

**Published:** 2024-12-17

**Authors:** Putri Permatasari, Haruka Goto, Manabu Miyamoto, Yasunori Oumi, Yogi Wibisono Budhi, Shigeyuki Uemiya

**Affiliations:** 1Department of Material Science and Processing, Gifu University, Gifu 501-1113, Japan; putri.permatasari.s4@s.gifu-u.ac.jp (P.P.); harukagoto821@gmail.com (H.G.); 2Department of Chemistry and Biomolecular Science, Gifu University, Gifu 501-1113, Japan; miyamoto.manabu.v5@f.gifu-u.ac.jp (M.M.); oumi.yasunori.x2@f.gifu-u.ac.jp (Y.O.); 3Department of Chemical Engineering, Institut Teknologi Bandung, Bandung 40116, Indonesia; y.wibisono@itb.ac.id

**Keywords:** simulation, CO_2_ methanation, NH_3_ decomposition, Pd membrane reactor, combined reaction, reactor performance

## Abstract

In a previous study, we developed an integrated reaction system combining NH_3_ decomposition and CO_2_ methanation within a membrane reactor, significantly enhancing reactor performance through efficient H_2_ separation. Ru/Ba/γ-Al_2_O_3_ and Ru/ZrO_2_ were employed as catalysts for each reaction. To ensure the accuracy and reliability of our results, they were validated through 1D models using FlexPDE Professional Version 7.21/W64 software. Key parameters such as reactor arrangement, catalyst bed positioning, overall heat transfer coefficient, rate constants, and H_2_ permeance were investigated to optimize system efficiency. The study revealed that positioning the NH_3_ decomposition on the shell side and CO_2_ methanation on the tube side resulted in a better performance. Additionally, shifting the methanation catalyst bed downward by approximately one-eighth (10 mm from 80 mm) achieves the highest CO_2_ conversion. A sensitivity analysis identified the rate constant of the NH_3_ decomposition catalyst and the H_2_ permeance of the membrane as the most influential factors in enhancing CO_2_ conversion. This highlights the priority of improving membrane H_2_ permeance and catalytic activity for NH_3_ decomposition to maximize system efficiency.

## 1. Introduction

The reliance of humans on fossil fuels has driven rapid industrial growth since the 18th century, but at a high environmental cost [1,2]. Among the issues, global warming attributed to carbon dioxide (CO_2_) constitutes 73.5% of the emissions and is the most critical, precipitating the alarming rise in temperatures and sea levels [3]. Corrective actions, such as altering land use, can reduce CO_2_ emissions by 40–70 ppm (10–30% of emissions), and natural solutions like reforestation help, but are insufficient alone due to water and crop yield limitations [4,5,6,7]. Several global initiatives have been implemented, including the Kyoto Protocol and the Paris Agreement [6,8]. However, despite these initiatives, emissions remain beyond projected levels, highlighting the urgent need for effective emission reduction technology [6,9,10].

Carbon Capture, Utilization, and Storage (CCUS) is increasingly recognized as an effective way to reduce CO_2_ emissions in various industries [9,11]. A notable advancement in this field is the methanation reaction, a procedure that employs CO_2_ and hydrogen (H_2_) to generate methane (CH_4_)—a versatile substance widely employed in various applications [9,12,13]. Methane has significant potential as a clean energy source for the future, especially when produced as green methane using renewable energy rather than fossil fuels [14,15,16]. The methanation process not only produces methane but also recycles CO_2_, thereby reducing carbon emissions [14,17]. For this approach to remain sustainable, the hydrogen used should ideally be green hydrogen derived from renewable energy sources.

Chemical transformations reliant on H_2_ face challenges due to costly H_2_ storage and transportation [9,10,18]. Methods using liquid H_2_, organic hydrides, and ammonia (NH_3_) are being explored [9,10,19,20]. NH_3_ stands out due to its H_2_ density and LNG-like liquefaction compatibility, meaning that the strategy involves using renewable energy to generate hydrogen, converting it into NH_3_ for storage and transport [9,21,22].

Hydrogen separation is essential to facilitate NH_3_’s role in hydrogen production [23]. Pd alloy-based inorganic membranes are highly efficient for hydrogen separation [24,25,26]. Electroless plating (ELP), the method used in this study, is the most common process, alongside chemical vapor deposition (CVD), physical vapor deposition (PVD), and electrodeposition (EPD) [27,28,29,30,31,32,33]. Membrane reactors play a crucial role in chemical process optimization by conducting reactions and separations simultaneously, improving both cost and energy efficiency [34,35]. The catalytic membrane method improves reaction selectivity and efficiency by manipulating thermodynamic equilibrium, isolating products, and enhancing purification and separation performance [35,36,37,38].

Significant progress has been made in NH_3_ decomposition using membrane reactors, where hydrogen removal enhances conversion rates and reaction kinetics due to hydrogen permeation [39,40]. We have demonstrated a combined system of NH_3_ decomposition and CO_2_ methanation using a membrane reactor [41]. The hydrogen separation membrane enhances ammonia decomposition by removing H_2_ from the system, and heat exchange between exothermic and endothermic reactions can save energy. Furthermore, using a membrane reactor to separate the two reactions increases the selectivity of the methanation reaction compared to mixing the two reactions in a single packed bed reactor, hence the use of a membrane reactor is favorable [41].

In system optimization, computational simulation is essential because it makes it possible to assess different operating conditions without the need for costly physical trials [42,43,44,45,46,47]. This covers sensitivity analyses and reaction kinetics computations. By combining catalyst and membrane data from previous research and employing computational methods to identify ideal settings, this work seeks to improve the combined system using a membrane reactor.

This research aims to address the urgent need to reduce greenhouse gas emissions and enhance hydrogen production efficiency. It concentrates on optimizing NH_3_ decomposition and CO_2_ methanation in a hydrogen-permeable membrane reactor. Efficiency is increased by the exothermic CO_2_ methanation process, which supplies energy for the endothermic NH_3_ decomposition. This study promotes scalable, affordable hydrogen production and aids in the fight against climate change by filling in gaps in the literature on catalyst performance, membrane separation, and heat transport and creating a simulation model to improve operations. This paper focuses on the computational analysis and simulation of a membrane reactor system for ammonia decomposition. Full experimental details regarding catalyst development, membrane fabrication, and system components can be found in our previous experimental study [41]. The current study builds on these experimental results to perform an in-depth computational analysis to optimize system performance and understand the reaction behavior.

## 2. Experimental Section

### 2.1. Determination of the Reaction Rate Formula

A Ru/Ba/γ-Al_2_O_3_ catalyst was used for the ammonia decomposition reaction. The exploration of NH_3_ decomposition involved a systematic derivation of activation energy (*E_A_*) and frequency factors (*k*_0_). At the outset, the focus centered on selecting an appropriate model for this reaction. The Temkin–Phyzev model (Equation (1)) was deemed optimal, owing to its ability to provide a holistic overview of the reaction’s trajectory. Drawing upon established reaction order values, where *α* is 1.410 and *β* is −1.176, a foundational basis was established for the analysis [48,49,50]. For ammonia decomposition, a reverse reaction was not considered because the presence of a hydrogen-permeable membrane prevents a reverse reaction from occurring.

Moving forward, an understanding of the reaction rate equation for CO_2_ methanation was also critical. For this reaction, a Ru/ZrO_2_ catalyst and Lunde and Kester’s model (Equation (2)) were used [51]. In this context, the consideration expanded to encompass the significance of the reverse reaction [52,53,54,55,56,57]. Reaction order values, where *n* is 0.85, were sourced from established studies [53]. Table 1 shows the experimental conditions used for determining the activation energy and frequency factor, as well as the experimental conditions used to determine the suitability of the reaction rate for the model. Figure 1 shows a schematic of the reaction equipment used in this system. 

### 2.2. The Arrangement of the Reactors

A preliminary experimental study was conducted to establish a benchmark. The reaction conditions are shown in Table 2 and the dimensions of the system can be seen in Figure 2. The main equations used in this study are shown in Equations (1)–(12). Because we have two reactions in one system, there are two possible arrangements, as seen in Figure 1. Arrangement 1: CO_2_ methanation in the tube, NH_3_ decomposition in the shell. Arrangement 2: NH_3_ decomposition in the tube, CO_2_ methanation in the shell. These configurations significantly affect the heat transfer in the reactor, which crucial for the combined reaction system’s efficiency. It is also important to note that the Pd layer is coated on the outer surface of the Al_2_O_3_ support. This point is crucial for determining the membrane’s area and the reactor’s diameter measurements for each point. Following that, our membrane uses an asymmetric support with a small pore size on the outside and a large pore size on the inside. This has implications for the simulation’s results. Assumptions included steady-state conditions, ideal gas behavior, and plug flow characteristics. The device’s dimensions are shown in Figure 2, with membrane and catalyst layer lengths of 80 mm.

FlexPDE Professional Version 7.21/W64 was used for calculations. A one-dimensional model was used, ignoring radial distribution and side reactions. Our results agree with Lundin et al.’s criteria for deciding whether to use 1D or 2D models in membrane reactors for hydrogen production [58]. They also assume the absence of axial and radial dispersion, which enhances the modeling technique for systems exhibiting such characteristics. The four key criteria considered are the Damköhler number (Da), Péclet number (Pe), transit parameter (θ), and equilibrium parameter (ε). In this study, the Péclet number (Pe) exceeds the critical Péclet number (Pe_crit_), which confirms the validity of the 1D modeling approach. The model does not explicitly incorporate axial heat conduction, but it remains valid and provides valuable insights into the reactor system’s behavior, as validated by experimental data. Physical property values used for simulation can be found in Table 3. The Table 4 presents the simulation conditions for the combined reaction within arrangements 1 and 2.

Reaction rate(1)$${-r}_{NH3}=k_{0,NH3}\;exp\left(\frac{-E_{A}}{RT}\right){{\times P}_{NH3}}^{\alpha }{P_{H2}}^{\beta }$$(2)$${-r}_{CO2}=k_{0,\;CO2}\;exp\left(\frac{-E_{A}}{RT}\right)\left\{{P_{CO2}}^{n}{P_{H2}}^{4n}-\frac{{P_{CH4}}^{n}{P_{H2O}}^{2n}}{{K_{eq}\left(T\right)}^{n}}\right\}$$
Material Balance
(3)$$\frac{dF_{NH3}}{dx}=A_{in}\rho_{cat}\left(-r_{NH3}\right)$$
(4)$$\frac{dF_{H2(in)}}{dx}=1.5A_{in}\rho_{cat}\left(r_{NH3}\right)-Qi$$
(5)$$\frac{dF_{CO2}}{dx}=A_{out}\rho_{cat}\left(-r_{CO2}\right)$$
(6)$$\frac{dF_{H2(out)}}{dx}=4A_{out}\rho_{cat}\left(-r_{CO2}\right)+Qi$$
Heat Balance


When ammonia decomposition is on the outside and carbon dioxide methanation is on the inside

(7)
$$\begin{array}{l} \left(F_{NH3}C_{pm,NH3}+F_{H2}C_{pm,H2}+F_{N2}C_{pm,N2}\right)\frac{dT_{gn}}{dx}=A_{in}\rho_{cat}\left(-r_{NH3}\right)\left({∆H}_{R,T}\right)+U_{1}{MA}_{1}\left(T_{gc}-T_{gn}\right)+\\ U_{2}{MA}_{2}\left(T_{g0}-T_{gn}\right) \end{array}$$


(8)
$$\begin{array}{l} \left(F_{CO2}C_{pm,CO2}+F_{H2}C_{pm,H2}+F_{CH4}C_{pm,CH4}+F_{H2O}C_{pm,H2O}\right)\frac{dT_{gc}}{dx}=A_{out}\rho_{cat}\left(-r_{CO2}\right)\left({∆H}_{R,T}\right)+\\ U_{1}{MA}_{1}\left(T_{gn}-T_{gc}\right)+{C_{pm}}_{H2}Qi\left(T_{gn}-T_{gc}\right) \end{array}$$



The overall heat transfer coefficient
(9)$$\frac{h_{w}d_{p}}{\lambda_{F}}=\left(\frac{d_{p}}{d}\right)\left(\frac{\lambda_{er}}{\lambda_{F}}\right)\left\{{a_{1}}^{2}+\frac{\Phi \left(b\right)}{\xi }\right\}$$

The total heat transfer coefficient through the Pd membrane (U_1_) [59] and the outer wall (U_2_) [60](10)$$\frac{1}{U_{1}Z{MA}_{1}}=\frac{1}{h_{in}Z{MA}_{1}}+\frac{{(d}_{2}-d_{1})/2}{\lambda_{al}A_{m1}}+\frac{1}{h_{out}Z{MA}_{2}}$$(11)$$\frac{1}{U_{2}Z{MA}_{3}}=\frac{1}{h_{out}Z{MA}_{3}}+\frac{{(d}_{4}-d_{3})/2}{\lambda_{sus}A_{m2}}$$
Hydrogen Permeation
(12)$$Qi=J\times {MA}_{1}\times \left({P_{H_{2}\left(in\right)}}^{0.5}-{P_{H_{2}(out)}}^{0.5}\right)$$

### 2.3. Examination of Catalyst Layer Position

We aimed to optimize catalyst layer placement in the reactor by shifting the CO_2_ methanation catalyst layer by *x* mm towards the gas outlet. The optimal arrangement is determined by comparing the total CO_2_ conversion rates to maximize CO_2_ utilization. Figure 1 (upper right) shows the settings used.

### 2.4. Parameter Change

We manipulated the reaction rate constant, hydrogen permeance, and total heat transfer coefficient to determine their impact on CO_2_ conversion. These variables were chosen because of their critical roles in regulating reaction kinetics, material transport, and energy motion within the system. The initial variables were multiplied by a factor ***a***, ranging from 0 to 25. The CO_2_ conversion rate was used to assess the effects of these changes. Equations (1), (2) and (10)–(12) show the modified variables in red.

## 3. Results and Discussion

### 3.1. Determination of the Reaction Rate Formula

Figure 3 presents the determination of (*E_A_*) and (*k*_0_) for both reactions. For NH_3_ decomposition, the Arrhenius plot analysis determined an *E_A_* of 60.5 kJ/mol and a *k*_0_ of 2.10 × 10^−3^ mol/g/s/Pa*^α+β^*. This value is in line with the literature which reports that the activation energy of the Ru/Al_2_O_3_ catalyst ranges from 60 to 90 kJ/mol, so even though it is lower, it is still considered valid for use in this study [59,61,62]. For CH_4_ methanation, the analysis yielded an *E_A_* of 100.4 kJ/mol and a *k*_0_ if 1.35 × 10^−16^ mol/g/s/Pa^5n^. Integrating these results into the established equations and comparing them with empirical data confirm that both formulas align with consistent trends, demonstrating their validity and applicability to the reaction kinetics.

### 3.2. The Arrangement of the Reactors

Two options exist for the combined reaction setup: methanation on the shell side with decomposition on the tube side, or vice versa (Figure 2). The choice significantly impacts the system’s behavior and outcomes, considering reaction kinetics, heat transfer, and efficiency. A preliminary experimental study was conducted to establish a basis for comparison and determine the practicality of the proposed configurations. The results revealed that arrangement 2 outperformed the other options (Figure 4(1)), supporting its potential for an improved performance. These findings add credibility to the subsequent simulation analyses and highlight the research’s promising trajectory.

The model results aligned with experimental findings, showing arrangement 2 to be superior. Both reactions’ conversion rates increased in this configuration (Figure 4(2)). This is due to NH_3_ decomposition being the rate-determining step and placing it on the shell side allowed for a larger catalyst volume and increased hydrogen production, enhancing CO_2_ methanation conversion. In addition, by placing the ammonia decomposition reaction, which is an endothermic reaction, closer to the heat source (in this case the reactor wall), the reaction will occur more easily so that the conversion in arrangement 2 is better than that in arrangement 1.

Based on the hydrogen pressure graph shown (Figure 4(3)), arrangement 2 shows a much higher hydrogen pressure on the NH_3_ side reaching about 2.8 kPa, while arrangement 1 has a more balanced pressure distribution with a value of about 1.2–1.3 kPa on the NH_3_ side and 0.5 kPa on the CO_2_ side. Both arrangements show a significant increase in pressure at the beginning of the reactor (0–20 mm) before reaching a stable condition after 40 mm, but arrangement 2 produces a higher-pressure gradient between the two sides of the reactor which can increase the hydrogen transfer rate through the Pd membrane. This indicates that arrangement 2 is more effective for producing hydrogen from NH_3_ decomposition, although the large pressure difference between the two sides can affect the efficiency of hydrogen transfer through the membrane, while arrangement 1 offers a more even pressure distribution which may be more beneficial for long-term operation stability.

The temperature profile graph in Figure 4(4) shows a clear difference between arrangement 1 and arrangement 2 during the combined reactions process. In arrangement 2, the temperature is higher, especially on the CO_2_ side, reaching approximately 676 K. In contrast, arrangement 1 has a lower and more uniform temperature profile across both sides of the reactor. Both arrangements show a sudden drop and spike in temperature near the reactor’s inlet (0–10 mm) before stabilizing. However, arrangement 2 creates a larger temperature difference between the NH_3_ and CO_2_ sides compared to arrangement 1. This demonstrates that the arrangement of reactions significantly affects the temperature distribution in the reactor.

Our analysis considers the heat transfer rate, temperature difference, thermal conductivity, and surface area. The heat transfer dynamics in arrangement 2 are shown in Figure 5. Negative values indicate heat release, while positive values indicate heat absorption. Upon closer examination, a notable trend occurs. Heat transfer through the outer reactor wall registers as being negative at approximately Z = 25 mm, indicating that heat is released from the shell side to the outer environment. This is intriguing considering that the decomposition reaction is endothermic which indicates that the methanation reaction on the tube side produces a significant amount of heat. This heat generation makes up for the heat absorbed by the decomposition reaction and produces extra amounts of heat that can be discharged through the reactor wall. Near the reactor’s entrance, a significant positive value emerges, possibly due to the initial interaction between NH_3_ and the catalyst, causing a rapid start to the endothermic reaction. This region has a lower temperature than other sections. CO_2_ methanation depends on H_2_ being produced by the NH_3_ decomposition. Even under outstanding conditions, compensating for the endothermic reaction’s heat loss is not possible. Adopting this arrangement reduces heating costs, facilitates a more energy-efficient process, and makes it promising for long-term operations.

### 3.3. Examination of Catalyst Layer Position

The conversion rate plot for the previous simulation (Figure 4(2)) shows lower CO_2_ conversion rates near the entrance due to a limited hydrogen transmission from the NH_3_ decomposition side. This suggests that the catalyst near the inlet was not optimally utilized, as reduced hydrogen availability limits its effectiveness. Proper catalyst utilization and hydrogen availability are crucial for reaction performance. Figure 6 shows the results after the catalyst layer has been displaced by *x* mm. The highest conversion rates are achieved when the CO_2_ methanation catalyst is shifted between 9 and 13 mm. With an 80 mm catalyst bed height, the best positioning involves moving the CO_2_ methanation catalyst downward by one-eighth of the total height, equivalent to a 10 mm displacement, producing optimal CO_2_ conversion rates within the existing reactor configuration.

### 3.4. Parameter Change

An exploration of parameter variations was carried out to investigate the impact of several parameters on the system. This experiment looks at the effect of multiplying different variable values by ***a*** (0–25) on CO_2_ conversion. The first stage of our investigation focused on the consequences of changes in the total heat transfer coefficient. In this regard, there are two separate heat transfer components: *U*_1_, which represents heat transfer through the Pd film, and *U*_2_, which represents heat transfer through the outer reactor wall. Figure 7 shows that neither form of heat transfer had a discernible impact on the CO_2_ conversion rate. It was determined that, within the confines of the current reactor configuration, changes in heat transfer have no discernible effect on the overall CO_2_ conversion rate. The change in the heat transfer coefficient did not significantly affect CO_2_ conversion, likely due to the predominance of other factors such as reaction kinetics and hydrogen permeation within the system. Additionally, the internal thermal balance between endothermic and exothermic reactions may contribute to the system’s stability against variations in external heat transfer. These findings highlight the critical importance of optimizing catalysts and membranes, while still accounting for heat transfer considerations in the design of large-scale reactors.

Figure 7 also shows the results for NH_3_ and CO_2_ conversion rates in response to variations in the reaction rate constant (*k*). An increase in the reaction rate constant has a clear consequence: increased conversion rates for both NH_3_ decomposition (*k*_1_) and CO_2_ methanation reactions (*k*_2_). It is important to remember that the decomposition reaction’s reaction rate affects the system more. The efficiency of the methanation reaction is dependent on the quantity of hydrogen generated during NH_3_ decomposition. Therefore, there is a bigger improvement in the conversion rate of the methanation reaction due to the rapid reaction rate in the decomposition phase greatly increasing the availability of hydrogen. This interaction highlights the complex relationship between reaction kinetics and reaction interdependencies within the system, shedding light on strategic opportunities for improving system performance through the careful adjustment of reaction rate constants.

The results of varying hydrogen permeance (*J*) levels can also be seen. Notably, increasing the membrane’s permeation capability accelerates the methanation reaction. This trend results from the direct relationship between membrane permeation and the kinetics of the H_2_-CO_2_ reaction. As membrane permeation increases, more hydrogen is available to interact with CO_2_, increasing the overall conversion rate. This finding emphasizes the importance of hydrogen permeance in influencing the reaction dynamics within the system, as well as the potential for increasing the efficiency of the combined reaction process.

We also examined temperature variations for each sensitivity analysis parameter. However, these data were excluded because no significant temperature changes were seen. These findings imply that changing the examined parameters has little effect on the temperature profile in the reactor system.

The results as seen in Figure 8 support a key trend: changing the rate constant for NH_3_ decomposition and increasing hydrogen permeance improves CO_2_ conversion more than changing the rate constant for CO_2_ methanation. This interesting observation shows that the CO_2_ methanation catalyst’s activity remains relatively high in its current configuration. The most important consideration is the critical roles of hydrogen production and permeance. Both factors emerge as significant determinants of the combined reaction system’s efficiency. This emphasizes the importance of future endeavors, with a primary focus on developing better NH_3_ decomposition catalysts and hydrogen permeation membranes. Increasing the efficiency of these critical components has the potential to significantly improve CO_2_ conversion rates and overall reaction system performance, ushering in a new era of increased reaction kinetics and operational effectiveness.

## 4. Conclusions

This study offers important insights into the pathways for optimizing the reaction system, pinpointing catalyst placement and reactions as well as important elements that enhance overall operating efficiency. It was found that the carbon dioxide methanation on the tube side and the decomposition process on the shell side gave better results than the contrary. Strategically relocating the CO_2_ methanation catalyst by about one-eighth of the catalyst layer’s height resulted in a noticeable improvement in the system’s efficiency. This result highlights how crucial catalyst positioning is to improve reaction kinetics. The greatest gains in efficiency were found when the rate constants for hydrogen permeance and NH_3_ decomposition were increased. A hydrogen-permeable membrane and a more active NH_3_ decomposition catalyst were shown to play crucial roles.

## Figures and Tables

**Figure 1 membranes-14-00273-f001:**
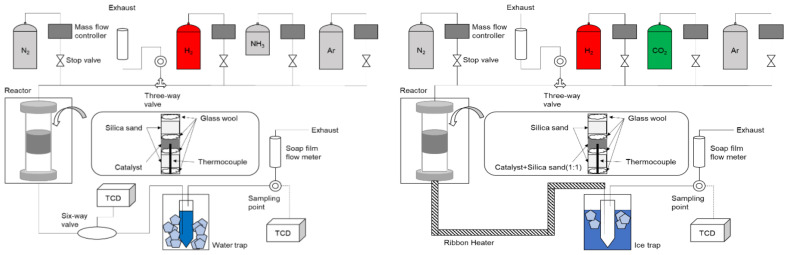
Schematic of reaction equipment used for NH_3_ decomposition (**left**) and CO_2_ methanation (**right**) reactor.

**Figure 2 membranes-14-00273-f002:**
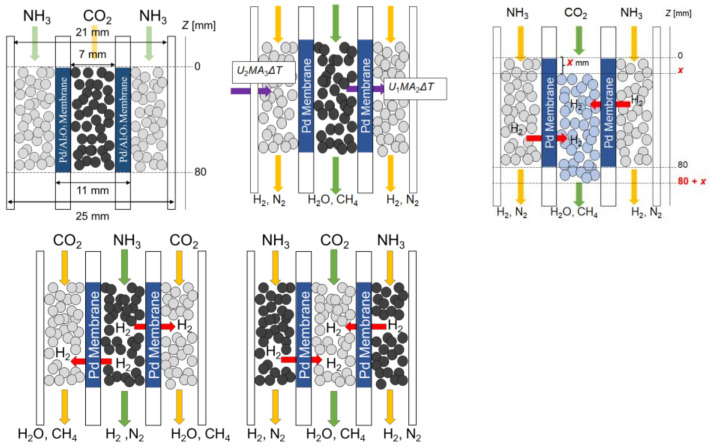
The device’s dimensions (**upper left**), heat transfer in the system (**upper middle**), the position of the catalyst layer (**upper right**), and the arrangement of the reactors: arrangement 1 (**bottom left**) and arrangement 2 (**bottom right**).

**Figure 3 membranes-14-00273-f003:**
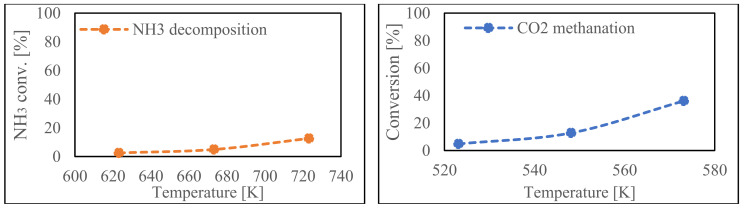
Catalyst performance (**top**) and comparison of experimental and simulation results (**middle**-**bottom**) for ammonia decomposition reaction (**left**) and carbon dioxide methanation (**right**).

**Figure 4 membranes-14-00273-f004:**
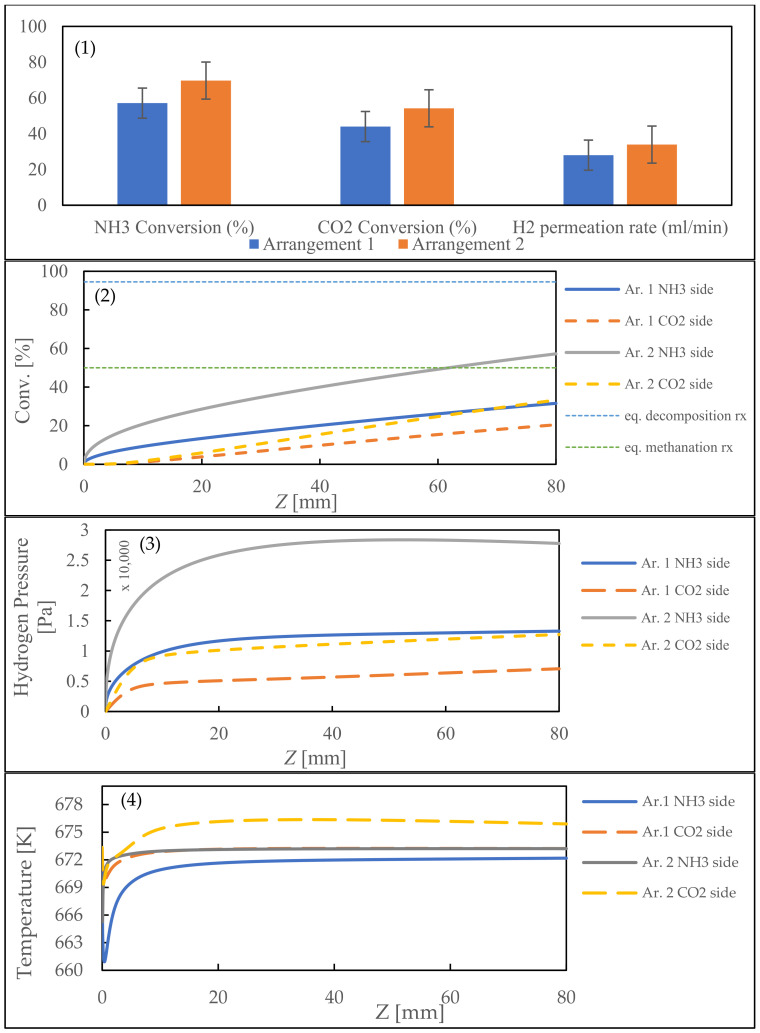
Experimental results (**1**) vs. simulated results for conversion rate (**2**), hydrogen pressure, (**3**) and temperature profile (**4**) obtained from arrangements 1 and 2.

**Figure 5 membranes-14-00273-f005:**
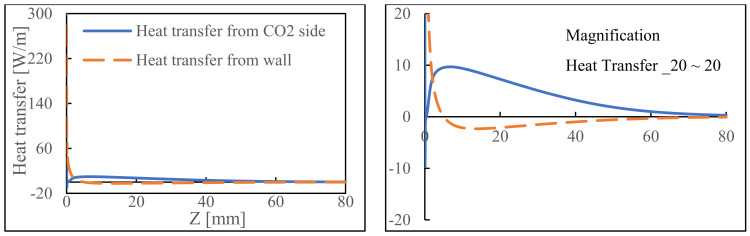
Heat transfer from arrangement 2 and its magnification.

**Figure 6 membranes-14-00273-f006:**
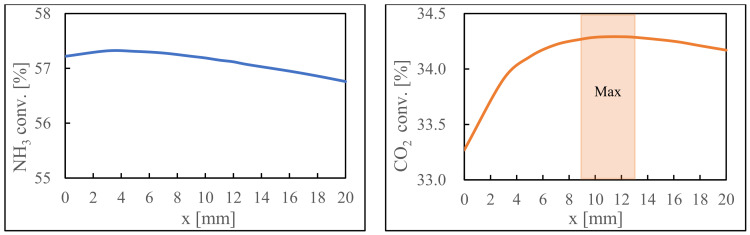
Conversion rates of both reactions when the CO_2_ methanation catalyst is moved.

**Figure 7 membranes-14-00273-f007:**
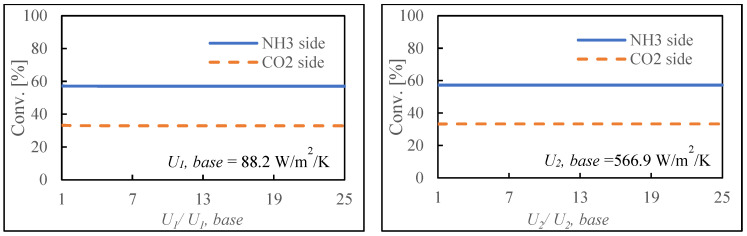
Effect of changes in heat transfer (**upper**), reaction rate constant (**middle**), and hydrogen permeance (**bottom**) on the system.

**Figure 8 membranes-14-00273-f008:**
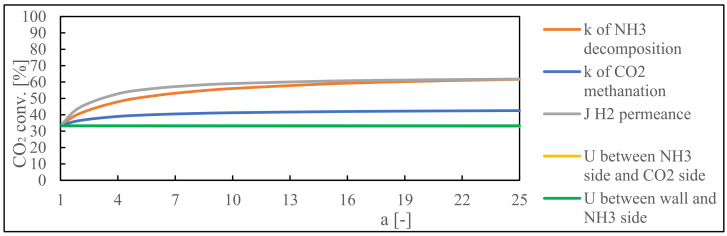
Sensitivity analysis of CO_2_ conversion (combined) where a (0–25) is the sensitivity analysis variable.

**Table 1 membranes-14-00273-t001:** Experimental conditions for determining reaction rate formula (in packed bed reactor).

	Catalyst Performance			Model Suitability (W/F)					Model Suitability (Temp.)					
Catalyst weight (g)	0.5			2					0.5					
NH_3_ flow rate (mL-stp/min)	40			145	67	30	23	16	40					
W/F (g.min/mol-_NH3_)	280			308	668	1486	1945	2859	280					
Temperature (K)	623.15	673.15	723.15	623.15					623.15		673.15		723.15	
Pressure (atm)	1			1					1					
Catalyst weight (g)	0.5			0.5					0.5					
CO_2_/H_2_ flow rate (mL-stp/min)	40/160			40/160	15/60	9/36	6/24	3/12	40/160					
W/F (g.min/mol-_CO2_)	280			289	721	1186	1772	3475	280					
Temperature (K)	523.15	548.15	573.15	573.15					523.15	548.15	573.15	598.15	623.15	673.15
Pressure (atm)	1			1					1					

**Table 2 membranes-14-00273-t002:** Experimental conditions to determine the arrangement of the reactor.

Inside catalyst weight (g)	1.5	**Arrangement 1**	
Outside catalyst weight (g)	10	heat transfer coefficients *U_1_* (W/m^2^/K)	113.7
NH_3_ flow rate (mL-stp/min)	40	heat transfer coefficients *U_2_* (W/m^2^/K)	160.6
CO_2_ flow rate (mL-stp/min)	15	**Arrangement 2**	
Reaction temperature (K)	623	heat transfer coefficients *U_1_* (W/m^2^/K)	88.2
		heat transfer coefficients *U_2_* (W/m^2^/K)	566.9

**Table 3 membranes-14-00273-t003:** Physical property values used for simulation.

*d*_1_ [m]	7.00 × 10^−3^		
*d*_2_ [m]	1.10 × 10^−2^		
*d*_3_ [m]	2.10 × 10^−2^		
*d*_4_ [m]	2.50 × 10^−2^		
*d*_p_ [m]	5.00 × 10^−4^		
*ε* [-]	4.00 × 10^−1^		
*ε*_w_ [-]	7.00 × 10^−1^		
*λ*_al_ [W/m/K]	3.60 × 10^1^		
*λsus* [W/m/K]	1.60 × 10^1^		
*λzr* [W/m/K]	4.00 × 10^0^		
**Physical property value**	**NH_3_**	**N_2_**	**H_2_**
*ai* [J/mol/K]	2.79 × 10^1^	3.12 × 10^1^	2.71 × 10^1^
*bi* [J/mol/K^2^]	2.38 × 10^−2^	−1.66 × 10^1^	9.27 × 10^−3^
*ci* [J/mol/K^3^]	1.80 × 10^−5^	2.68 × 10^−5^	−1.88 × 10^1^
*di* [J/mol/K^4^]	−1.99 × 10^1^	−1.97 × 10^1^	7.65 × 10^−9^
*ΔH*_f,298K_ [J/mol]	−4.71 × 10^3^	0.00 × 10^0^	0.00 × 10^0^
*M*_w_ [g/mol]	1.70 × 10^1^	2.80 × 10^1^	2.02 × 10^0^
*K* [Pas/K^0.5^]	1.80 × 10^−6^	1.38 × 10^−6^	6.71 × 10^−7^
*C* [K]	6.26 × 10^2^	1.03 × 10^2^	8.30 × 10^1^
*μ_i, 673 K_* [Pas]			
*ρ_0,_i* [kg/m^3^]	7.71 × 10^−1^	1.25 × 10^0^	8.98 × 10^−2^
**Physical property value**	**CO_2_**	**CH_4_**	**H_2_O**
*ai* [J/mol/K]	1.98 × 10^1^	1.93 × 10^1^	3.22 × 10^1^
*bi* [J/mol/K^2^]	7.34 × 10^−2^	5.21 × 10^−2^	1.92 × 10^−3^
*ci* [J/mol/K^3^]	−6.10 × 10^1^	1.20 × 10^−5^	1.06 × 10^−5^
*di* [J/mol/K^4^]	1.72 × 10^−8^	−1.93 × 10^1^	3.60 × 10^−9^
*ΔH*_f,298K_ [J/mol]	−4.06 × 10^4^	−7.71 × 10^3^	−2.49 × 10^4^
*M*_w_ [g/mol]	4.40 × 10^1^	1.60 × 10^1^	1.80 × 10^1^
*K* [Pas/K^0.5^]	1.66 × 10^−6^	1.08 × 10^−6^	
*C* [K]	2.74 × 10^2^	1.98 × 10^2^	
*μ_i, 673 K_* [Pas]			2.45 × 10^−5^
*ρ_0_,i* [kg/m^3^]	1.98 × 10^0^	7.17 × 10^−1^	3.48 × 10^−1^

**Table 4 membranes-14-00273-t004:** Simulation condition of combined reaction inside of arrangement 1 and 2.

SELECT		SELECT	
ngrid	1.00 × 10^0^	ngrid	1.00 × 10^0^
errlim	1.00 × 10^−5^	errlim	1.00 × 10^−5^
cell_limit	1.00 × 10^2^	cell_limit	1.00 × 10^2^
VARIABLES		VARIABLES	
*F* _NH3_	1.00 × 10^−9^	*F* _NH3_	1.00 × 10^−9^
*F* _H2(NH3)_	1.00 × 10^−9^	*F* _H2(NH3)_	1.00 × 10^−9^
*F* _CO2_	1.00 × 10^−9^	*F* _CO2_	1.00 × 10^−9^
*F* _H2(CO2)_	1.00 × 10^−9^	*F* _H2(CO2)_	1.00 × 10^−9^
*T* _gc_	1.00 × 10^−9^	*T* _gc_	1.00 × 10^−9^
*T* _gn_	1.00 × 10^−9^	*T* _gn_	1.00 × 10^−9^
DEFINITIONS		DEFINITIONS	
*Z* [m]	8.00 × 10^−2^	*Z* [m]	8.00 × 10^−2^
*f*_NH3_ [ml-stp/min]	4.00 × 10^1^	*f*_NH3_ [ml-stp/min]	4.00 × 10^1^
*f*_CO2_ [ml-stp/min]	1.50 × 10^1^	*f*_CO2_ [ml-stp/min]	1.50 × 10^1^
*W*_in_ [g]	1.50 × 10^0^	*W*_in_ [g]	1.50 × 10^0^
*W*_out_ [g]	1.00 × 10^1^	*W*_out_ [g]	1.00 × 10^1^
*T*_g0_ [K]	6.73 × 10^2^	*T*_g0_ [K]	6.73 × 10^2^
*U*_1_ [W/m^2^/K]	1.14 × 10^2^	*U*_1_ [W/m^2^/K]	8.82 × 10^1^
*U*_2_ [W/m^2^/K]	1.61 × 10^2^	*U*_2_ [W/m^2^/K]	5.67 × 10^2^
*J* [mol/m^2^/s/Pa^0.5^] [41]	1.71 × 10^−4^	*J* [mol/m^2^/s/Pa^0.5^] [41]	1.71 × 10^−4^
INITIAL VALUE		INITIAL VALUE	
*F* _NH3_	*F* _NH3,0_	*F* _NH3_	*F* _NH3,0_
*F* _H2(NH3)_	*F* _H2(NH3),0_	*F* _H2(NH3)_	*F* _H2(NH3),0_
*F* _CO2_	*F* _CO2,0_	*F* _CO2_	*F* _CO2,0_
*F* _H2(CO2)_	*F* _H2(CO2),0_	*F* _H2(CO2)_	*F* _H2(CO2),0_
*T* _gc_	*T* _g0_	*T* _gc_	*T* _g0_
*T* _gn_	*T* _g0_	*T* _gn_	*T* _g0_

## Data Availability

The data will be made available upon request.

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
