# Peer review of "Combined Reaction System for NH_3_ Decomposition and CO_2_ Methanation Using Hydrogen Permeable Membrane Reactor in 1D Model Analysis"

_membranes, 2024, doi:10.3390/membranes14120273_

Round 1

Reviewer 1 Report

Comments and Suggestions for Authors

The work: »Combined Reaction System of NH3 Decomposition and CO2 Methanation using Hydrogen Permeable Membrane Reactor in 1D Model Analysis« by Permatasari et al. Is about modelling of membrane system of two coupled reactions. The work is sound, combining simplest models to understand behaviour, however the research depth and contribution to the scientific fields can be still increased by a large factor and major revision is advised.

-The conversions are low, why not increasing the temperature of NH3 cracking? Why not vary the parameters to achieve full conversions?

-Based on sensitivity analysis, NH3 cracking is the limiting reaction. The »arrangment 2« seems to be better, because of higher volume of NH3 cracking catalyst, although this is not mentioned in the paper.

-Please, provide an exact literature:  (line 98) Reaction order values, ո is 0.85, were sourced from established studies

-Please provide a total pressures in table 1

-In heat balance equations, there is no heat flow for H2 permeation on NH3 cracking side. What is the reason?

-Also from the equation, I could assume that NH3 cracking is on the outside and CO2 methanation in on inside. Please add in text that is clear.

(The image cannot be displayed, please see the attachment.)

Provide more discussion in the introduction, especially on the reason for conversion of NH3 to produce methane (which is typically cheaper) and on utilization and modelling of membrane-based operations (https://www.sciencedirect.com/science/article/pii/S0959652624019280)

Reviewer 2 Report

Comments and Suggestions for Authors

The scope of the paper is to examine optimal configurations for conducting ammonia decomposition and CO2 methanation in an integrated membrane reactor. The experimental details are completely missing. The mathematical model presents a few basic concerns concerning its validity. The sensitivity analysis is limited and has very little general importance in my opinion.  

In the intro, this sentence is not clear to me:"In this system, the hydrogen separation membrane facilitates both reactions, with membrane-assisted hydrogen separation increasing methane selectivity due to active hydrogen species on the Pd membrane surface". It is not clear why the methane selectivity will increase, and compared to which other species? CO ?

Is it justified to ignore RWGS reaction? Please explain.   II

66What is the reason the rate of reaction 1 is independent on N2 partial pressure?

Reaction 1 is endothermic and reaction 2 is exothermic. Then, why the temperature used to study the kinetics of reaction 1 is HIGHER than the temperature of reaction 2 (figure 2)? Shouldn't the exothermic reaction be carried at higher temperature to enable heat transfer between the reactors? 

The authors claim that "It is also important to note that the Pd layer is coated on the outer surface of the Al₂O₃ support." Please explain why it is important.

Section 2.3 title:Examination of Catalyst Layer Movement. The layer doesn't move. I suggest to call it "|ayer position".

In table 2, temperature of 350 K, what is this temperature?  Feed? External?

It is not clear what is the source of the data points in figure 2? Were these measured by the authors as part of this work or taken from the literature? If these were measured, please describe the full experimental system. Furthermore, this figure does NOT show an Arrhenius plot as the authors claim. Please revise the figure.

In line 175 the authors claim that the arrangement significantly affects the system's dynamics, but they are analyzing it using a steady state model, not a dynamic one.  Thus, their claim is inappropriate and is not supported.

Experimental details are completely missing; what are the details of the various units of the system? How was the membrane made? How was the reactor sealed? etc.

In line 191 the authors claim that the model considers thermal conductivity, but that is not true based on equations 7-9. They actually NEED to consider axial heat conduction (dispersion) which is an important term when simulating exothermic reaction in fixed beds, to enable the prediction of moving fronts.

In figure 3 please compare the measured and simulated conversion on the same axis.

Temperature profiles in the reactors are not presented. Please add and analyze. 

Another comment related to the one above, in figure 6, it is very troubling that the authors show that the model results are completely insensitive to 2 main parameters, the heat transfer coefficients. This warrants revising the model. The authors should better explain this result.

Changing the kinetic rate is not a simple goal. Alternatively, would increasing the reactor length achieve the same result? That is (one of) the proper design parameters to consider in reactor design, rather than the kinetic rate constant which is given.

Comments on the Quality of English Language

A few typos have been found, please review the entire manuscript for typos and grammar. For example, "use" should be "used" in line 88, "dan" and "shows" instead of "show" in line 155 etc.

Other than a few typos, some sentences are cumbersome and not clear.

Round 2

Reviewer 1 Report

Comments and Suggestions for Authors

I don't have further comments. I recommend publication of the paper.